# Biological Characteristics and Odontogenic Differentiation Effects of Calcium Silicate-Based Pulp Capping Materials

**DOI:** 10.3390/ma14164661

**Published:** 2021-08-18

**Authors:** Yemi Kim, Donghee Lee, Hye-Min Kim, Minjoo Kye, Sin-Young Kim

**Affiliations:** 1Department of Conservative Dentistry, College of Medicine, Ewha Womans University, Seoul 07986, Korea; yemis@ewha.ac.kr; 2College of Medicine, The Catholic University of Korea, Seoul 06591, Korea; dong524@naver.com; 3Department of Conservative Dentistry, Seoul St. Mary’s Hospital, College of Medicine, The Catholic University of Korea, Seoul 06591, Korea; hmtoto@naver.com (H.-M.K.); judy796@naver.com (M.K.)

**Keywords:** biological properties, calcium silicate-based cement, direct pulp capping materials, odontogenic differentiation

## Abstract

We compared calcium silicate-based pulp capping materials to conventional calcium hydroxide in terms of their biological properties and potential effects on odontogenic differentiation in human dental pulp stem cells (hDPSCs). We cultured hDPSCs on disks (7-mm diameter, 4-mm high) of ProRoot MTA (Dentsply Tulsa Dental Specialties), Biodentine (Septodont), TheraCal LC (Bisco), or Dycal (Dentsply Tulsa Dental Specialties). Cell viability was assessed with cell counting (CCK) and scanning electron microscopy (SEM). Odontogenic activity was assessed by measuring alkaline phosphatase (ALP) activity and gene expression (quantitative real-time PCR). CCK assays showed that Dycal reduced cell viability compared to the other materials (*p* < 0.05). SEM showed low and absent cell attachment on TheraCal LC and Dycal disks, respectively. hDPSCs exposed to TheraCal LC and Dycal showed higher ALP activity on day 6 than the control group (*p* < 0.05). The day-9 *Runx2* expression was higher in the ProRoot MTA and TheraCal LC groups than in the control group (*p* < 0.05). On day 14, the ProRoot MTA group showed the highest dentin sialophosphoprotein levels (not significant; *p* > 0.05). In conclusion, various pulp capping materials, except Dycal, exhibited biological properties favorable to hDPSC viability. ProRoot MTA and TheraCal LC promoted higher *Runx2* expression than Biodentine. Future studies should explore the odontogenic potential of pulp capping materials.

## 1. Introduction

Direct pulp capping is a common treatment for dental pulp that is exposed in pin-point sized holes created when removing dental caries or repairing clinical crown fractures [1,2]. The two main purposes of direct pulp capping treatments are to restore the dentin-pulp complex and to avoid root canal treatment [1].

Calcium hydroxide has been used in direct pulp capping for a long time, and it is thought to be the conventional material for pulp capping treatments [3]. Calcium hydroxide is alkaline, and it discharges Ca^2+^ and OH^−^. The Ca ions form apatite crystals, which stimulate mineralization. The OH ions have antibacterial properties. When calcium hydroxide is placed on exposed pulp, a superficial necrotic zone is formed. Subsequently, the mineralization process occurs directly against the necrotic area [3]. Notably, replacing calcium hydroxide with Dycal (Dentsply Tulsa Dental, Johnson City, TN, USA) has some disadvantages. The highly soluble Dycal triggers inflammatory processes that support tunnel formation, which allows bacterial intrusion [4].

Mineral trioxide aggregate (MTA) was introduced in dentistry because of its ability to create a seal and its biocompatibility. Moreover, it displayed odontoblastic and cementoblastic activities [5,6]. In previous reports, MTA reliably provided good clinical outcomes more frequently than calcium hydroxide, in direct pulp capping applications [7,8,9,10]. In a clinical study, the success rate of MTA was close to 80% after a median of 42 months; in contrast, calcium hydroxide showed a success rate of 59% [8]. However, MTA has some disadvantages: it requires a prolonged setting time, it is difficult to handle, and it is prone to discoloration. Therefore, calcium silicate-based cements were introduced. One calcium silicate-based cement, Biodentine (Septodont, Saint-Maur-des-Fossés, France), required shorter setting times, displayed less discoloration, and showed more favorable clinical results than MTA [11,12,13]; therefore, Biodentine was thought to be preferable to MTA as a pulp capping material.

Recently, resin-modified calcium silicate (TheraCal LC, Bisco, Schamberg, IL, USA) was introduced, due to its enhanced material properties. Among its several advantages, it can be immediately cured with light exposure, it displays high mechanical strength, it is easy to handle, and it can be applied precisely onto the exposed pulp [14]. Resin-modified calcium contains a hydrophilic monomer, and the levels of Ca^2+^ and OH^−^ discharged are higher than those discharged by ProRoot MTA (Dentsply Tulsa Dental Specialties, Tulsa, OK, USA) and Dycal [15]. Consequently, resin-modified calcium exhibited better antibacterial properties than MTA, and it stimulated mineralized nodule formation more efficiently than MTA. TheraCal LC is manufactured in a viscous solution that can flow over a surface, before it is cured with light. Therefore, it can be easily placed in coronal teeth, if forms a tight seal, and restorations can be finished in a single day. However, TheraCal LC was reported to be more cytotoxic than ProRoot MTA and Biodentine in human dental pulp stem cells (hDPSCs) [16,17].

The present study aimed to compare the biological properties and odontogenic differentiation potentials of calcium silicate-based pulp capping materials to those of conventional calcium hydroxide in hDPSC cultures.

## 2. Materials and Methods

### 2.1. Cell Preparation

The Institutional Review Board of the Catholic University of Korea approved the study protocol (IRB No. KC19SNSI0086). The hDPSC line was established at Top Cell Bio, Inc (Seoul, Korea). The experiments were performed with hDPSCs that had been passaged 4 times, as described previously [18]. Cells were grown in HyClone Minimum Essential Medium (α-MEM; GE Healthcare Life Sciences, Pittsburgh, PA, USA) supplemented with HyClone 10% fetal bovine serum (GE Healthcare Life Sciences), 100 U/mL penicillin, and 100 μg/mL streptomycin. Cell cultures were maintained at 37 °C in a humified atmosphere with 5% CO_2_. In colony-forming tests, the majority of hDPSCs maintained a spindle-shaped morphology, which is typical of mesenchymal stem cells. All experimental procedures were performed under aseptic conditions.

### 2.2. Preparation of Various Direct Pulp Capping Material Eluates

We tested the following four direct pulp-capping materials: ProRoot MTA (Dentsply Tulsa Dental Specialties), Biodentine (Septodont), TheraCal LC (Bisco), and Dycal (Dentsply Tulsa Dental Specialties). Table 1 shows the components of each material [19]. All materials were prepared according to manufacturer instructions. We created material disks that were 7 mm in diameter and 4 mm in height with sterile rubber molds in an aseptic atmosphere. All disks were stored in a humidified atmosphere at 37 °C for 2 days. Then, they were released in growth medium for one week at a concentration of 5 mg/mL. These eluates were used in subsequent experiments to test the effects of the different materials on cell biological properties.

### 2.3. Cell Proliferation Evaluation

To evaluate the effects of the four direct pulp-capping materials on cell proliferation, we assessed cell numbers with cell-counting kit-8 (CCK-8; Dojindo Molecular Technologies Inc., Kumamoto, Japan). Briefly, hDPSCs were seeded in growth medium, with or without a material eluate, in 24-well cell culture plates (SPL Life Sciences, Pocheon, Korea) at a density of 1.0 × 10^4^ cells/well. The proliferation rates of hDPSCs cultured with the different pulp-capping material eluates were evaluated over 6 days. The control group included hDPSCs cultured without an experimental material eluate. According to the CCK-8 kit instructions, we added WST-8 [2-(2-methoxy-4-nitrophenyl)-3-(4-nitrophenyl)-5-(2,4-disulfophenyl)-2H tetrazolium, monosodium salt] and incubated the hDPSCs for 45 min at 37 °C. Then, we determined the optical density at 450 nm with a Power Wave XS spectrometer (BioTek Instruments, Winooski, VT, USA). Each group was analyzed in sextuplicate. This assay depended on the capability of mitochondrial dehydrogenases to reduce WST-8, and thereby produce formazan dye.

### 2.4. Cell Morphology Evaluation

We performed scanning electron microscopy (SEM) to evaluate the morphology of hDPSCs after they were placed in direct contact with disks made of the test materials. Briefly, cells were seeded at a density of 5.0 × 10^4^ cells/well onto each type of material disk. In the control group, cells were seeded on glass coverslips (SPL Life Sciences Co., Ltd., Korea). After 3 days of direct contact, hDPSCs cultured on the test materials were fixed with 2 mL of 2% buffered paraformaldehyde (Biosesang, Seongnam, Korea) for 4 h. SEM was performed as described previously [20].

### 2.5. Cell Viability Evaluation

hDPSCs were seeded at a density of 1.0 × 10^4^ cells/well and incubated with eluates of the various pulp capping materials for 5 days. The control group included hDPSCs cultured without an experimental material eluate. On days 3 and 5, the cells were double-stained to detect live (*green*) and dead (*red*) cells with a LIVE/DEAD™ Cell Imaging Kit (488/570; Molecular Probes, Life Technologies, CA, USA). We performed qualitative analyses of cell viability with an inverted microscope and digital image processing software (ZEN 2012, AxioVision; Carl Zeiss Microscopy, Jena, Germany).

### 2.6. Alkaline Phosphatase Enzyme Activity Evaluation

The tested disks were released in osteogenic medium for one week at a concentration of 5 mg/mL. These eluates were used in subsequent experiments to test whether the different materials affected odontogenic differentiation. On days 3 and 6, we evaluated intracellular alkaline phosphatase (ALP) activity in hDPSCs with the Senso-Lyte^®^ p-nitrophenylphosphate (pNPP) alkaline phosphatase assay kit (AnaSpec, Fremont, CA, USA), as described previously [18]. Briefly, cells were fixed in 4% paraformaldehyde (Biosesang) for 1–2 min, then rinsed with TBST buffer (1X Tris-Buffered Saline, 0.05% Tween 20 Detergent). Cells in each well were stained with the solution provided in the alkaline phosphatase detection kit (Millipore, Darmstadt, Germany). Then, stained cells were stored in the dark at room temperature for 15 min. We determined the optical density at 405 nm with a Power Wave XS spectrometer (BioTek Instruments). Each group was evaluated in sextuplicate.

### 2.7. Quantitative Real-Time Polymerase Chain Reaction Evaluation of Odontogenic Potential

Cells in experimental groups were harvested at 9 days to analyze the expression of genes related to odontogenic differentiation, including runt-related transcription factor 2 (*Runx2*), osteocalcin (*OCN*), and osteopontin (*OPN*). A parallel set of cells in each group was harvested at 14 days to analyze the expression of dentin matrix protein-1 (*DMP-1*) and dentin sialophosphoprotein (*DSPP*). Briefly, total RNA was isolated from hDPSCs with Tri Reagent (TR118; Molecular Research Center, Inc., Cincinnati, OH, USA). The RNA concentration was determined, and 500 ng of total RNA was used as template. We performed quantitative real-time polymerase chain reactions (qPCR) with the iTaq Universal SYBR Green One-Step Kit (BioRad, Hercules, CA, USA) and the CFX96 Real-Time PCR detection system, equipped with Bio-Rad CFX manager software, version 3.1 (BioRad). The qPCR protocol was: 50 °C for 10 min; 95 °C for 1 min, 45 cycles of 95 °C for 10 s, 60 °C for 30 s; then 60 cycles of 60 °C for 5 s, with increases of + 0.5 °C per cycle for a melting curve analysis. Primers were designed in GenBank (Table 2). Amplified gene levels were normalized to the GAPDH expression level, and relative expression was measured as the fold change, compared to expression in the control group.

### 2.8. Statistical Analyses

Statistical analyses were performed with SPSS ver. 24.0 (IBM Corp., Armonk, NY, USA). Data distributions were evaluated with the Shapiro–Wilk normality test. We conducted repeated measures analysis of variance to evaluate CCK and ALP results. We performed a one-way analysis of variance and Tukey post hoc test to compare experimental and control groups on day 6. The qPCR results were analyzed with the Kruskal–Wallis test and Mann–Whitney analysis. P-values less than 0.05 were considered statistically significant.

## 3. Results

### 3.1. Cell Proliferation Evaluation

The levels of cell proliferation were not significantly different among the ProRoot MTA, Biodentine, TheraCal LC, and control groups at any experimental time-point (*p* > 0.05). Cells exposed to Dycal exhibited the lowest proliferation rate after 24 h (Figure 1). Notably, on day 6, cell proliferation was significantly different between the Dycal and control groups (*p* < 0.05).

### 3.2. Cell Morphology Evaluation

Our SEM evaluation showed that hDPSCs formed a well-attached cell layer on the ProRoot MTA and Biodentine disks, similar to the control group. In contrast, we observed poor cell attachment on the TheraCal LC disks. In the Dycal group, dead cells were observed on the experimental disks (Figure 2).

### 3.3. Cell Viability Evaluation

The viability assay revealed live cells in the ProRoot MTA and Biodentine groups, similar to the control group. Notably, the TheraCal LC and Dycal groups had lower numbers of live cells and higher numbers of dead cells compared to the control group (Figure 3).

### 3.4. ALP Enzyme Activity Evaluation

On day 6, hDPSCs incubated with TheraCal LC and Dycal showed significantly higher ALP activity than cells incubated with ProRoot MTA or Biodentine (*p* < 0.05; Figure 4). On day 6, ALP staining was significantly higher in the TheraCal LC group compared to ALP staining in the other experimental groups (Figure 5).

### 3.5. qPCR Evaluation of Odontogenic Potential

Figure 6 summarizes the effects of test materials on osteogenic gene expression after 9 days. *Runx2* expression was significantly upregulated in hDPSCs incubated with ProRoot MTA and TheraCal LC (*p* < 0.05). *DMP-1* expression was upregulated in all experimental groups on day 9; however, these upregulated levels did not significantly differ from the expression level observed in the control group (*p* > 0.05, Figure 7). *DPSS* was upregulated in hDPSCs incubated with ProRoot MTA on day 14; however, this upregulation did not significantly differ from the expression level observed in the control group (*p* > 0.05, Figure 7).

## 4. Discussion

When dental pulp is exposed, direct pulp-capping is required to maintain pulp vitality and to promote the development of reparative dentine. Materials used for direct pulp capping must be noncytotoxic to hDPSCs and facilitate odontogenic differentiation. Moreover, quick-setting materials enable one-visit restorative treatments. In this study, we evaluated the biological properties of calcium silicate-based pulp capping materials and their potential effects on odontogenic differentiation, compared to the known effects of conventional calcium hydroxide treatment.

Our CCK, SEM, and live/dead staining analyses showed that, compared to Dycal, ProRoot MTA and Biodentine promoted cell proliferation and attachment. The effect of TheraCal LC on cell proliferation was similar to the effect observed in the control group, but better than that of Dycal. In the Dycal group, we observed apoptotic cells in the SEM evaluation, and increased dead cells in the viability staining evaluation, consistent with previous studies [21,22].

In previous studies, TheraCal LC exhibited high toxicity to DPSCs, due to its resin components, which induced the release of high levels of interleukin-8 (IL-8) and IL-6 [23,24]. These proinflammatory cytokines can lead to pain, and they interfere with direct pulp capping. Consequently, one author previously concluded that TheraCal LC should not be recommended for exposed pulp applications [24]. In that study, the TheraCal LC composition included a hydrophilic resin monomer, a hydrophobic resin monomer, and a hydrophilic filler, in addition to the Portland cement powder [19,25]. Methacrylate resin constituents have a negative effect on cells because they alter the lipid bilayer of phospholipid-containing cell membranes [19]. In contrast, in the present study, we used TheraCal LC eluate, which was made by incubating the TheraCal LC disk in osteogenic medium for one week. This eluate may have contained a lower concentration of the toxic components of TheraCal LC, compared to the concentrations studied in previous reports. This difference might explain the favorable cell response to TheraCal LC observed in our study.

In a previous randomized clinical trial, direct pulp capping with Biodentine showed 100% success (i.e., no failures) during a 12-month follow-up. In contrast, calcium hydroxide exhibited an accumulated failure rate of 13.64% [10]. Lipski et al., reported that Biodentine had an 82% success rate at 1–1.5 years after direct pulp capping, and the success rate was 90.0% among patients under 40 years of age [26]. Biodentine was associated with higher cell proliferation and migration rates than the rates observed with other pulp-capping materials [18,27,28]. The effects of Biodentine on dentin bridge formation were similar to those observed with ProRoot MTA [29,30]. These features contributed to the high success rates of Biodentine in direct pulp-capping procedures.

In the present study, we analyzed odontogenic differentiation potential in hDPSCs incubated with ProRoot MTA, Biodentine, TheraCal LC, and Dycal, based on ALP activity and osteogenic gene expression. We found that hDPSCs incubated with TheraCal LC showed higher ALP activity than hDPSCs incubated with ProRoot MTA and Biodentine on day 6 (Figure 4 and Figure 5). Moreover, ProRoot MTA and TheraCal LC were associated with higher *Runx2* expression than Biodentine (Figure 6). These results contrasted with findings from previous studies, which showed that Biodentine exhibited greater osteogenic potential than TheraCal LC [17,23,31].

Giraud et al., found that odontogenic markers, such as Nestin and DSP, were more highly expressed in a Biodentine group compared to a TheraCal LC group [23]. Moreover, a complete dentin bridge was found with Biodentine [23,31], and the TheraCal LC group exhibited only a scattered mineralized nodule in disordered pulp [23]. Another previous study investigated the human pulp reaction to a partial pulpotomy procedure with TheraCal LC. In that study, most patients showed the formation of an intermittent dentin bridge [17]. In the present study, we found that, compared to Biodentine, hDPSCs incubated with TheraCal LC showed higher ALP activity on day 6, and higher expression levels of *Runx2* and *OCN* on day 9. These differences might be explained by the higher release of OH^−^ and Ca^2+^ from TheraCal LC compared to ProRoot MTA and Dycal [15]. Therefore, dentin bridge formation may be influenced by factors other than the release of calcium ions. Accordingly, the favorable ALP activity and Runx2 expression associated with TheraCal LC in this study may not be related to the ability to form a dentin bridge.

Biodentine causes calcium hydroxide formation and the release of OH^−^ and Ca^2+^ ions. Calcium ions stimulate hDPSCs. This stimulation facilitates cell differentiation and increases calcium nodule formation, which contributes to reparative dentin bridge formation. On the other hand, TheraCal LC does not cause calcium hydroxide formation, as shown in X-ray diffraction analyses [25]. This failure might be due to a lack of sufficient moisture for the appropriate hydration of the tricalcium silicate components in TheraCal [25]. Calcium phosphate, and the subsequent apatite formation in a wet environment, is recommended to stimulate DPSC differentiation, odontogenesis, and cementogenesis [32]. Silicon ions can also contribute to mineralization. Their release from pulp capping materials reportedly facilitates young bone formation by stimulating osteoblasts [33]. Accordingly, the silicon ion components in Biodentine appear to facilitate dentin bridge formation [23].

Biodentine can also support regenerative processes by increasing the secretion of transforming growth factor β1 (TGF-β1) and fibroblast growth factor 2 (FGF-2). These effects are not observed with TheraCal LC [23]. Secretion of TGF-β1 from DPSCs may stimulate regenerative dentine formation, specifically in young, immature teeth with advanced caries [34]. In dental pulp, TGF-β1 facilitates stem cell migration and odontoblast differentiation [35,36]. Laurent et al., reported that Biodentine meaningfully increased TGF-β1 release from DPSCs [37]. No previous studies have assessed TGF-β1 and FGF-2 secretion, in detail, with the use of TheraCal LC. Therefore, further studies are necessary to elucidate those effects.

It has been reported that half of the methacrylate monomer double bonds in resin polymers remain unresponsive. However, when these bonds contact dental pulp, they exert cytotoxic effects on pulp fibroblasts [38,39]. Additionally, it has been shown that nontoxic concentrations of monomers can suppress the expression of dentin sialoproteins and osteonectin [40]. Both these proteins are related to dentin bridge formation; therefore, arresting their expression may explain the reduced formation of a dentin matrix in procedures that used TheraCal LC, compared to procedures that used MTA or Biodentine. However, in the present study, hDPSCs incubated with ProRoot MTA or TheraCal LC exhibited higher *Runx2* expression on day 9 than hDPSCs incubated with Biodentine. Furthermore, *DMP-1* and *DSPP* gene expression levels on day 14 were higher in hDPSCs incubated with ProRoot MTA compared to hDPSCs incubated with Biodentine or TheraCal LC (Figure 7). Further investigations in an animal model are needed to evaluate mRNA and protein expression over the long term.

The present study had several limitations. First, we did not confirm our histological evaluations in an animal model. It is possible that ALP activity and *Runx2* expression are higher in hDPSCs incubated with TheraCal LC, compared to hDPSCs incubated with Biodentine, at the cellular level, but these results could be different at the tissue level, where dispersed dentin bridges were observed with TheraCal LC. Second, we used the various pulp capping materials in the set state. We seeded hDPSCs directly onto the experimental materials after they dried for the SEM evaluation, and we used experimental material eluates for the CCK assay. These conditions are likely to explain why the toxicity of TheraCal LC was not as high in this study as it was in previous studies. Further animal studies are needed to examine the biological response and the effects on odontogenic differentiation when direct pulp capping procedures are performed with Biodentine and TheraCal LC.

## 5. Conclusions

In summary, we found that ProRoot MTA and Biodentine showed better biological properties than Dycal. Compared to the control group, hDPSCs exposed to TheraCal LC exhibited similar cell proliferation, but less cell attachment. We also observed higher ALP activity and *Runx2* expression in hDPSCs incubated with TheraCal LC compared to hDPSCs incubated with Biodentine. Overall, our findings indicated that ProRoot MTA, Biodentine, and TheraCal LC could be applied as substitute pulp-capping materials. These materials were associated with more favorable biological responses than Dycal. Further odontogenic differentiation studies are needed to compare the effects of Biodentine and TheraCal LC.

## Figures and Tables

**Figure 1 materials-14-04661-f001:**
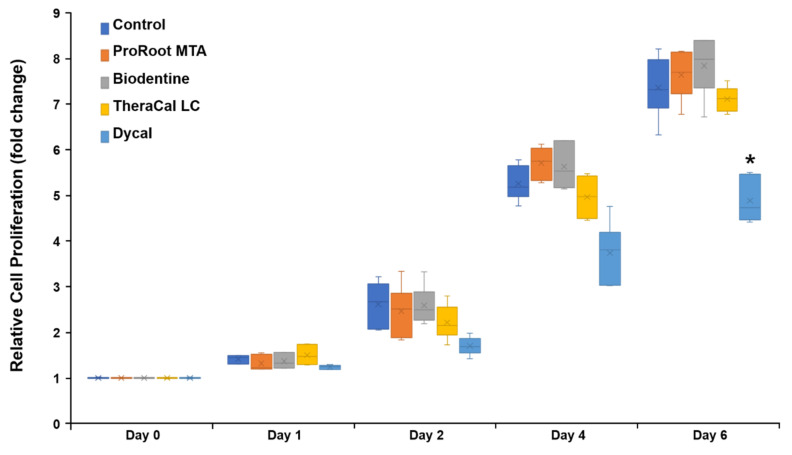
Relative proliferation of hDPSCs incubated with various pulp-capping materials, based on the cell-counting kit-8 (CCK) assay. * Statistically significant difference between the Dycal and control groups.

**Figure 2 materials-14-04661-f002:**
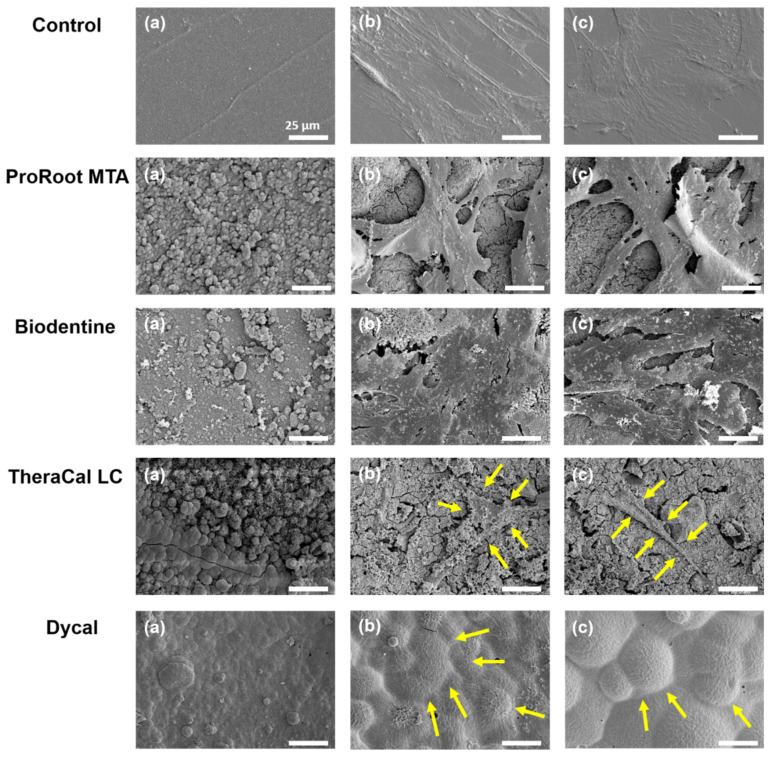
Scanning electron microscopy images show hDPSCs incubated with various pulp-capping materials. hDPSCs were cultured directly on top of solid material disks. (**a**) Material disks before adding hDPSCs (×250). (**b**) hDPSCs attached on each experimental disk (×250). (**c**) hDPSCs attached on each experimental disk (×1000) (scale bar = 25 μm). Arrows indicate poor cell attachment on TheraCal LC disk and dead cells on Dycal disk.

**Figure 3 materials-14-04661-f003:**
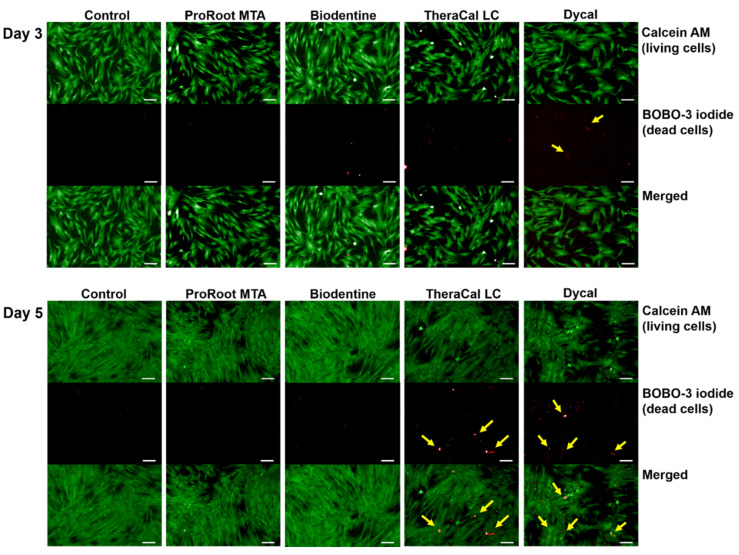
Representative images of cell viability in the presence of various pulp capping materials. Cells were double-stained with LIVE/DEAD cell stain. Upper panels show live hDPSCs (*green*); middle panels show dead hDPSCs (*red*); lower panels show superimposed images (scale bar = 200 μm). Arrows indicate dead cells in the TheraCal LC and Dycal groups.

**Figure 4 materials-14-04661-f004:**
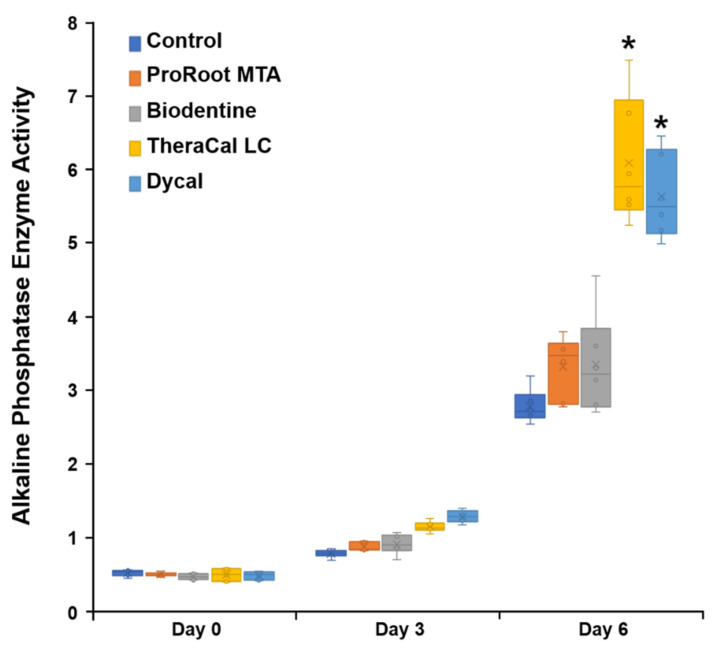
Alkaline phosphatase (ALP) enzyme activity in hDPSCs incubated with various pulp capping materials. * Statistically significant difference between experimental and control groups.

**Figure 5 materials-14-04661-f005:**
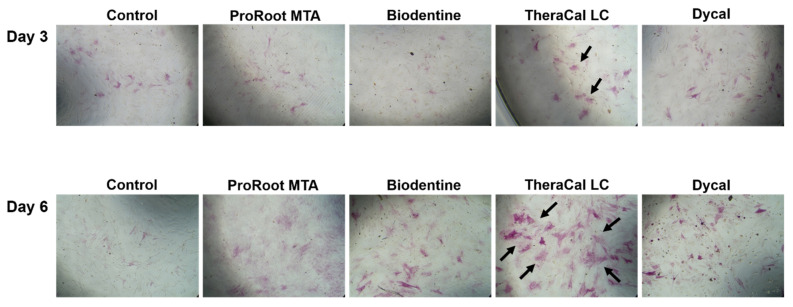
Representative images of alkaline phosphatase (ALP) staining in hDPSCs incubated with eluates of various pulp capping materials. Arrows indicate high ALP staining in the TheraCal LC group.

**Figure 6 materials-14-04661-f006:**
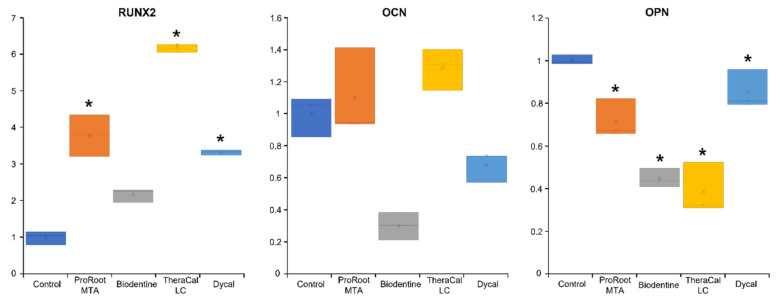
Relative expression levels of osteogenic genes, *Runx2*, *OCN*, and *OPN*, in hDPSCs incubated for 9 days with various pulp capping materials. * Statistically significant difference between experimental and control groups.

**Figure 7 materials-14-04661-f007:**
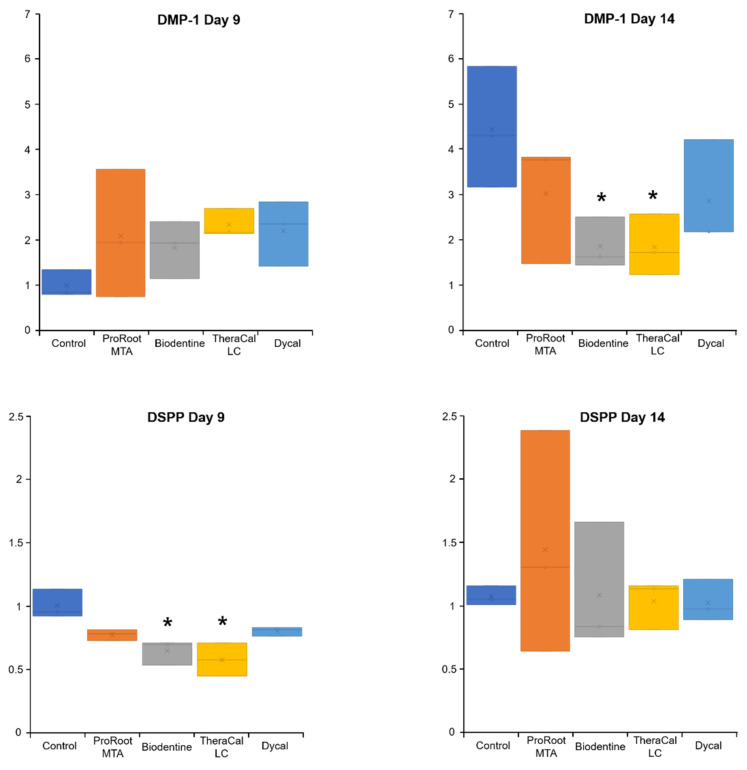
Relative expression levels of osteogenic genes, *DMP-1* and *DSPP*, in hDPSCs incubated for (*left*) 9 and (*right*) 14 days with various pulp-capping materials. * Statistically significant difference between experimental and control groups.

**Table 1 materials-14-04661-t001:** Manufacturers and chemical components of the experimental materials for direct pulp capping.

Product	Manufacturer	Components	Lot Number
ProRoot MTA	Dentsply Tulsa Dental Specialties, Tulsa, OK, USA	Portland cement (tricalcium silicate, dicalcium silicate, and tricalcium aluminate) 75%, calcium sulfate dihydrate (gypsum) 5%, Bismuth oxide 20%.	0000186484
Biodentine	Septodont, Saint- Maur-des-Fossés, France	Tricalcium silicate, dicalcium silicate, calcium carbonate, calcium oxide, and zirconium oxide, in powder form. Water, calcium chloride, and soluble polymer, in aqueous liquid form.	B24553
TheraCal LC	Bisco, Schamberg, IL, USA,	Portland cement (calcium silicates), fumed silica, Bis-GMA, polyglycol dimethacrylate.	1900004558
Dycal	Dentsply Tulsa Dental, Johnson City, TN, USA	Base paste: calcium phosphate, calcium tungstate, zinc oxide, iron oxide pigments, 1,3-butylene glycol disalicylate. Catalyst paste: calcium hydroxide, zinc oxide, zinc stearate, titanium oxide, iron oxide pigments, N-ethyl-o/p-toluene sulphonamide.	160801

**Table 2 materials-14-04661-t002:** Primer sequences for evaluating osteogenic gene expression in hDPSCs.

Gene	Primer Sequence
runt-related transcription factor 2 (*Runx2*)	Forward 5′-AAG TGC GGT GCA AAC TTT CT-3′ Reverse 5′-TCT CGG TGG CTG CTA GTG A-3′
osteocalcin (*OCN*)	Forward 5′-GTG CAG AGT CCA GCA AAG GT-3′
	Reverse 5′-TCA GCC AAC TCG TCA CAG TC-3′
osteopontin (*OPN*)	Forward 5′-TGA AAC GAG TCA GCT GGA TG-3′
	Reverse 5′-TGA AAT TCA TGG CTG TGG AA-3
dentin matrix protein-1 (*DMP-1*)	Forward 5′-TGG TCC CAG CAG TGA GTC CA-3′
	Reverse 5′-TGT GTG CGA GCT GTC CTC CT-3′
dentin sialophosphoprotein (*DSPP*)	Forward 5′-GGG AAT ATT GAG GGC TGG AA-3′
	Reverse 5′-TCA TTG TGA CCT GCA TCG CC-3′
GAPDH	Forward 5′-TGT CAT CAA CGG GAA GCC-3′
	Reverse 5′-TTG TCA TGG ATG ACC TTG-3′

## Data Availability

The datasets used and/or analyzed during the current study are available from the corresponding author on reasonable request.

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
