# Peer review of "Biological Characteristics and Odontogenic Differentiation Effects of Calcium Silicate-Based Pulp Capping Materials"

_materials, 2021, doi:10.3390/ma14164661_

Round 1

Reviewer 1 Report

Article presents an interesting and up-to-date issue. It is well written. However, I have some suggestions.

Please check grammar, punctuation and spelling thought the text.

Introduction

Please elaborate on clinical background of the study, why this research is important, what are complications?

Material and methods

Please add description of the control group for all methods.

Results

Figure 2,3,5 – describe in detail, add also arrows on the figure to indicate your observations.

Figure 1,4,6,7 – please present data as box and whiskers plots

Please divide results into subsection as they are divided in method section.

Discussion

Is there any link of obtained results to the composition of tested materials?. Please elaborate in detail. You might find some useful information in:

Kunert, M et al. Bio-Inductive Materials in Direct and Indirect Pulp Capping-A Review Article. Materials. 2020, 13, 5, 1204. DOI10.3390/ma13051204

Author Response

Please see the attachment PDF file.

Reviewer 2 Report

The reviewed manuscript has practical relevance, yielding interesting applications, worth publishing, once the authors address some aspects, which in my opinion require their attention.

Although the article is overall well-argued and the corresponding authors are well published, there are some parts of the manuscript which require attention, both as to its structure and English. I mention some indicative linguistic errors:

  • There name of the first author is misspelled, I suppose the surname is Kim not Kim1.
  • Page 1, Line 30, “Direct pulp capping procedure is commonly treated…” please check the syntax, the treatment surely does not refer to the procedure!
  • Page 1, Line 32, “The main purposes of direct pulp capping treatment are…” should either read “The main purpose of direct pulp capping treatment is…” or “The main purposes of direct pulp capping treatments are…”
  • Page 1, Line 34, “…long time which…” should read “…long time and…”
  • Page 1, Line 34, “…thought to…” should read “…thought as…”
  • Page 1, Line 36, “…and stimulate mineralization…” should read “…and stimulates mineralization…”
  • Page 1, Line 37, “…pulp, superficial…” should read “…pulp, a superficial…”

There are more of these syntax/linguistic issues throughout the text and I’d ask the authors to carefully proofread their manuscript again and consult a colleague versed in scientific English writing.

Although the authors do mention a reference for the Cell Preparation, I would ask them to summarize the process here, as to ease the evaluation of the procedure.

In chapter 2.4, the authors mention “Cells were seeded at a density of 5.0 104 cells/well”. Please elaborate what “5.0 104” refers to, is this a typo? Do you mean 5.0x104? Please check that all superscripts are correctly represented throughout the manuscript (e.g. page 4, line 104).

The title of chapter 2.5 (Live/dead Staining Evaluation), should be phrased more appropriately, maybe “Viability” is a better term than “Live/dead”.

Please maintain the same color-coding throughout all figures, e.g. all materials in figures 6 and 7 have the same color, in contrast to the bar diagrams of figures 1 and 4 which are easier to read.

Author Response

Please see the attachment PDF file.

Round 2

Reviewer 1 Report

The article was improved according to the comments. I have one additional suggestion:

Lines 191-192

‘Notably, the TheraCal LC group had lower numbers of live cells and higher numbers of dead cells’

Higher/lower than what? Other materials?

Congratulations on your article.

Reviewer 2 Report

The authors addressed most of my concerns.

Although they mention having the manuscript proof read by an English editing service, this is not reflected in the highlighted parts of the manuscript. It might be helpful to do so as to ease any linguistic concerns.
